# Effectiveness of Vestibular Rehabilitation for Patients with Degenerative Cerebellar Ataxia: A Retrospective Cohort Study

**DOI:** 10.3390/brainsci13111520

**Published:** 2023-10-28

**Authors:** Lisa L. Heusel-Gillig, Courtney D. Hall

**Affiliations:** 1Emory Dizziness and Balance Center, Emory University, Atlanta GA 30329, USA; 2James H. Quillen Veterans Affairs Medical Center, Mountain Home, TN 37684, USA; 3Physical Therapy Program, East Tennessee State University, Johnson City, TN 37614, USA

**Keywords:** cerebellar ataxia, vestibular rehabilitation, balance, gait, habituation exercises

## Abstract

Many patients with cerebellar ataxia have dizziness caused by oculomotor or peripheral vestibular deficits; however, there is little evidence supporting the use of vestibular rehabilitation for this population. The purpose of this study was to determine whether patients with degenerative cerebellar diseases improve following rehabilitation including vestibular exercises. A secondary aim was to identify variables associated with the outcomes. A retrospective chart review identified 42 ambulatory patients (23 men and 19 women; mean age = 54.5 ± 14.4 years) with cerebellar degeneration. Fourteen patients had ataxia only, twenty had ataxia and oculomotor abnormalities, and eight had ataxia with oculomotor and peripheral vestibular deficits. Patients received customized physical therapy including balance and gait training, as well as gaze stabilization and habituation exercises for vestibular hypofunction and motion-provoked dizziness. Primary outcome measures (Activities-specific Balance Confidence Scale, Tinetti Performance Oriented Mobility Assessment, Dynamic Gait index, and Sensory Organization Test) were evaluated at baseline and discharge. Patients improved (*p* < 0.05) on all outcome measures. Patients with vestibular deficits were seen for more visits compared to those with gait ataxia only (7.1 vs. 4.8 visits). This study provides evidence that patients with degenerative cerebellar disease improve in balance confidence, fall risk and sensory integration with therapy that includes vestibular rehabilitation.

## 1. Introduction

The degenerative cerebellar ataxias are a heterogeneous group of disorders resulting in progressive atrophy which is primarily of the cerebellum, but also may include brainstem and/or spinal pathways [1,2]. Individuals with cerebellar degeneration frequently experience dysarthria, dysmetria, tremors, abnormal oculomotor signs, dizziness, and gait ataxia [1,2,3,4]. Variants of this disorder that involve vestibular pathways may result in central and peripheral vestibular dysfunction, resulting in dizziness [5,6,7,8,9].

Historically, individuals with progressive cerebellar degeneration were not expected to improve significantly with rehabilitation. Physical therapy (PT) was provided for range of motion exercises to prevent contractures as well as instruction for the proper use of “orthoses, sticks, and strollers” to complete activities of daily living [1,10]. Subsequently, treatment interventions for ataxia frequently included Frenkel’s coordination exercises and proprioceptive neuromuscular facilitation techniques [11]. These limited therapeutic approaches have been called into question as recent studies have demonstrated improved coordination and postural control in patients with cerebellar ataxia using a variety of therapeutic interventions.

Recent systematic reviews have examined the current clinical practice of individuals with degenerative cerebellar ataxia. He et al. found 33 studies focusing on balance and coordination interventions with patients with genetic degenerative ataxia [12]. This systematic review identified evidence for improvements with traditional physical and occupational therapy, as well as intensive coordination training, core stability exercises, aerobic training, sensory integration, head-position based tongue electro-tactile biofeedback, proprioceptive stabilizer devices, virtual reality/video games, tango/dance, and tai chi. Another systematic review identified 14 studies producing moderate to high quality evidence. Although the studies had small sample sizes, improvements in ataxia severity, gait speed, and balance were shown following therapeutic interventions [13]. The review noted that a limitation of the current evidence for this population was the lack of a wide range of supervision, dosage, intensity, and types of treatments that made generalization difficult. Milne et al. included 17 studies with 292 participants having a variety of genetic degenerative ataxias [14]. Fifteen of the studies demonstrated significant improvements in at least one outcome measure following PT. The interventions included coordination exercises, balance training, cycling, treadmill training, and respiratory muscle training [14].

None of these more recent systematic reviews included vestibular rehabilitation as an intervention [12,13,14]. An older systematic review included two studies (one case report and one retrospective cohort study) that included vestibular exercises as a component of therapy [11]. The retrospective cohort study included patients with central vestibular disorders, of which a subgroup of 11 patients had cerebellar dysfunction, including stroke and degeneration [15]. While patients with central vestibular disorders as a group showed improvements in all outcome measures, the cerebellar dysfunction sub-group did not show significant improvements in any of the measures [15]. Most studies including subjects with degenerative cerebellar ataxia have focused on PT treatments for coordination, balance, and gait training, even though oculomotor signs and vestibular dysfunction are common impairments in this population [16,17,18,19,20]. Many patients with degenerative cerebellar ataxia who present to PT clinics with gait and balance deficits also have dizziness caused by central oculomotor deficits (e.g., smooth pursuit, abnormal saccades, visually enhanced vestibulo-oculo reflex, VVOR, and down beat nystagmus) or peripheral vestibular impairments (e.g., decreased vestibulo-ocular reflex, VOR). However, there is little evidence supporting the use of vestibular rehabilitation for this population.

The purpose of this study was to address this gap in the knowledge by determining: (1) whether patients with degenerative cerebellar ataxia can improve their balance confidence, fall risk, and sensory integration of balance with a customized plan of care including vestibular gaze stabilization and habituation exercises to address central or peripheral vestibular deficits; and (2) whether variables including age and time from onset of gait instability affect their rehabilitation outcomes.

## 2. Materials and Methods

### 2.1. Participants

This study was a retrospective cohort study of patients diagnosed with degenerative cerebellar ataxia who were referred for PT during a six-year period at a specialized dizziness and balance outpatient clinic. Inclusion criteria included completion of at least two sessions consisting of an initial evaluation, intervention, and discharge evaluation. Patients were excluded if they had cerebellar ataxia from strokes or tumors. The Emory University’s Institutional Review Board approved the study and the participants gave informed consent for use of their clinical data (IRB Study33-2002, 1191-2005). Patients were evaluated by neurologists specializing in movement disorders and/or vestibular disorders. All patients underwent vestibular nystagmography (VNG) for positional and oculomotor testing. Abnormalities in this cohort included spontaneous down beat nystagmus, direction-changing gaze-evoked nystagmus, saccadic smooth pursuit, and VOR cancellation deficits, as well as hypermetric and slow saccades. Peripheral vestibular deficits were identified with abnormal bithermal caloric or rotary chair testing. Caloric results of >25% asymmetry were considered diagnostic for unilateral vestibular hypofunction and results of <10 deg/s total response indicated bilateral vestibular hypofunction. Rotary chair testing confirmed bilateral loss in three subjects. Patients were divided into three groups for analysis: ataxia only (without oculomotor or vestibular impairments), ataxia with oculomotor deficits, and ataxia with both oculomotor and vestibular abnormalities.

The patient demographics, including age, gender, time from onset of ataxia, and vestibular/oculomotor abnormalities, were extracted from medical records. All participants were independent household ambulators with or without a cane or walker. Based on the gait and stance categories of the SARA ataxia scale, most subjects were in the 1–2 range, indicating mild to moderate ataxia. Balance confidence, fall risk, and sensory integration were evaluated during initial and discharge PT visits by the first author (LHG). The presence of peripheral neuropathy was noted but not assessed statistically because it was documented in only three subjects.

Patients were provided with a customized home exercise program (HEP) during their initial evaluation and participated in weekly 60 min supervised PT sessions ranging from 2 to 11 visits. Compliance with the home exercise program and therapeutic interventions were recorded in the medical records.

### 2.2. Measurement of Outcome Measures

Balance confidence was measured using the ABC scale, which is a self-reported assessment of confidence across a continuum of 16 activities (e.g., walking in the house, on stairs, on a ramp) [21]. Confidence was rated on a scale from 0 to 100%, with higher scores reflecting less fear of falling and a greater confidence in balance, and an overall average was calculated. A normal score was demonstrated by an overall average of at least 80% and the minimal clinically significant difference was a change of at least 10 points. The ABC has good test–retest reliability [21].

Fall risk was measured using the Dynamic Gait Index (DGI) or the Performance-Oriented Mobility Assessment (POMA). The DGI determines an individual’s ability to safely modify their gait in the presence of external demands (e.g., changing speed, turning their head, avoiding obstacles, and climbing stairs). The maximum total score possible is 24 and a total score of ≤19 indicates fall risk [22]. A meaningful change is defined as a change in score of at least 3 points. The DGI has excellent interrater, as well as test–retest, reliability and validity [23]. The POMA was developed by Tinetti for older community-dwelling adults and is a tool used widely to evaluate both balance and gait [24]. The version used in this study has a maximum score of 37 points (25 for the balance portion, and 12 for the gait portion) and a score <32 indicates fall risk. This version contains more demanding items than the original POMA such as feet together with eyes open and closed, as well as single leg stance [24]. The POMA-T (total balance and gait score) and POMA-B (balance only) have adequate reliability and validity for assessing mobility in older adults. The minimum detectable change for the original version is 5 points [25].

Sensory integration was measured using the computerized Neurocom Equitest Sensory Organization Test (SOT), which assesses the ability to utilize appropriate sensory input for balance while standing on a platform. The SOT is organized into a series of six conditions to assess the ability to utilize visual, somatosensory, and vestibular inputs [26]. Postural sway within each condition is estimated and a composite score (a weighted average of the 6 sensory conditions) is calculated. The composite score has good validity and reliability [26].

### 2.3. Interventions

#### 2.3.1. Balance and Gait Training

All patients participated in individualized balance exercises designed to improve static and dynamic postural stability and sensory integration. The static balance exercises included maintaining balance with altering vision and somatosensory cues as well as varied support bases. The dynamic balance exercises included activities involving voluntary weight shifts with visual feedback, such as the Wii Fit, to promote the appropriate use of ankle, hip, and stepping strategies. Gait activities included negotiating uneven terrains and obstacles, head turns, varied speeds, and unpredictable starts and stops. A customized daily HEP including coordination, sit to stand transfers without the use of arms for strengthening, and the balance activities mentioned above, in addition to walking for endurance, was prescribed and progressed as appropriate.

#### 2.3.2. Oculomotor Exercises

To improve their scanning of the environment, patients with VOR cancellation deficits were prescribed daily VOR X0 exercises, which involved following a target with their eyes and head moving together horizontally. Subjects with downbeat nystagmus causing dizziness, postural instability or double vision in eccentric gaze were taught behavioral strategies to locate their nystagmus null points. These strategies, including tilting the head or turning the head and eyes together to look peripherally, helped improve visual acuity [8,27].

#### 2.3.3. Vestibular Gaze Stabilization Exercises

Individuals diagnosed with a peripheral vestibular deficit (n = 8) who complained of visual blurring and dizziness with head movements performed gaze stabilization exercises, including VOR X1 viewing and gaze shifting between two targets [28]. The VOR X1 viewing exercise involved looking at a target (a letter) and turning the head as quickly as possible while maintaining focus on the target. Patients performed this exercise with both horizontal and vertical head turns, at far-distance (6–10 feet) and near-distance (arms-length) targets. The gaze shifting exercise involved alternately shifting the eyes and then head between two targets while keeping the target in focus. This activity was performed both horizontally and vertically. Each of the vestibular gaze stabilization exercises was performed for a duration of 1 min 3–5 times/day, for a total of 12–20 min, as recommended by the vestibular hypofunction clinical practice guidelines [28].

#### 2.3.4. Habituation Exercises

Patients with motion-provoked dizziness were prescribed vestibular habituation exercises with the goal of reducing their response to a noxious stimulus or movement, by exposing the individual incrementally to the provocative movement or environment repeatedly [29]. Participants who reported motion-provoked dizziness were evaluated using the Motion Sensitivity Test, which includes 10 items such as horizontal and vertical head turns, bending over, and 180-degree turns while standing [29]. Movements that provoked mild to moderate symptoms were identified and prescribed as part of a HEP to be performed 2 times daily. Participants were instructed to perform the movements at a speed to provoke mild to moderate symptoms, as well as to allow the symptoms to return to baseline levels prior to the next repetition or set. The number of repetitions and sets was gradually increased as tolerated. No more than 3–5 different movements were performed at a time.

### 2.4. Data Analysis

The data were summarized using descriptive statistics. To determine the impact of vestibular rehabilitation, 2 × 2 repeated-measures analyses of variance (RM ANOVA) were performed with time (pre and post treatment) as the ‘within subjects’ variable and group (Ataxia Only, Ataxia + Oculomotor, Ataxia + Oculomotor + Vestibular) as the ‘between subjects’ variable. The dependent variables included ABC, DGI, POMA, and SOT. To determine the impact of two variables on rehabilitation outcomes, 2 × 2 (group by time) RM ANOVAs were performed with age group (<65 years and ≥65 years) and time from onset (≤5 years and >5 years), as the ‘between subjects’ variables. The dependent variables included ABC, DGI, POMA, and SOT. The significance level was set at alpha = 0.05.

## 3. Results

The retrospective chart review identified 42 patients who were included in the data analyses. The mean age was 54 ± 14.4 years (range: 18–78) and 55% were male (Table 1). Fourteen patients (32%) had ataxia, but no oculomotor or VOR deficits (Ataxia Only); 20 patients (49%) had ataxia and central oculomotor deficits (smooth pursuit, saccades, spontaneous/direction-changing gaze-evoked nystagmus or VOR cancellation) based on VNG findings, but normal peripheral vestibular function (Ataxia + Oculomotor); while 8 subjects (19%) had ataxia, central oculomotor, as well as peripheral vestibular deficits (Ataxia + Oculomotor + Vestibular). Motion-provoked dizziness was a subjective complaint of the patients with oculomotor and vestibular impairments. There were no significant differences in baseline demographics (age, gender, number of comorbidities, years since onset) among the groups (*p* > 0.05), with the exception of the number of PT visits (*p* = 0.043). Post hoc analysis revealed that the Ataxia + Oculomotor + Vestibular group received more PT sessions than the Ataxia Only group (*p* = 0.01). Canes or walkers were used by 11 (26%) patients.

Following vestibular rehabilitation, all patients improved on all outcome measures (*p* < 0.001; Table 2). There were no significant interactions (*p* > 0.05) between time (pre- and post-treatment) and group (Ataxia Only, Ataxia + Oculomotor, Ataxia + Oculomotor + Vestibular) for the outcome measures, with the exception of DGI (*p* = 0.03). Post hoc analysis of DGI revealed that the Ataxia Only, Ataxia + Oculomotor, and Ataxia + Oculomotor + Vestibular groups were significantly different at baseline (*p* = 0.03), but not at discharge (*p* = 0.57), indicating that the Ataxia + Oculomotor + Vestibular group improved to a greater extent than the Ataxia + Oculomotor group.

To examine the impacts of different variables (age and time from onset), we examined the interaction effect of these factors on the impact of time (pre and post treatment). There were no significant interactions (*p* > 0.05) with the exception of age group (Table 3). There was one significant interaction (*p* = 0.04) between age group and rehabilitation, such that older patients had a significantly lower baseline POMA score (*p* = 0.02), but no difference at discharge (*p* = 0.66), which suggests greater improvements than the younger patients.

## 4. Discussion

The findings of this study are important because they indicate that vestibular rehabilitation may enhance traditional therapeutic interventions in individuals with degenerative cerebellar ataxia with vestibular involvement causing dizziness or visual blurring. Additionally, the outcomes were not affected by age, time from onset, or the presence of central or peripheral vestibular deficits. Other studies on patients with degenerative cerebellar ataxia have shown improvements in similar outcome measures to those used in our study, but dizziness was not addressed [12,13,14]. Most prospective controlled studies supporting vestibular rehabilitation examined people with peripheral vestibular hypofunction, not those with central vestibular or degenerative cerebellar disorders [30,31,32].

Although research focusing specifically on subjects with cerebellar dysfunction with dizziness is scarce, we did find studies of heterogeneous groups of subjects with central vestibular disorders that included participants with cerebellar degeneration [15,33,34]. A retrospective study of 48 participants with either central or peripheral vestibular disorders, utilizing a customized exercise program, including balance and gait training, sensory integration, and VOR gaze stabilization exercises, showed improvements in all diagnostic groups; however, the 11 patients with cerebellar dysfunction, including cerebellar stroke and cerebellar degeneration, improved the least [15]. The two outcome measures used, DGI and ABC, were the same as in our study. In the work by Brown et al., the mean DGI of the cerebellar subgroup (n = 8) changed minimally (by 1.5 points with a pre-intervention score of <12/24), as did the ABC (n = 10; changed by 5.9 points from an average of 44% to 49%). Compared to our study, they had a small sample size, and their cerebellar subjects started at lower scores on DGI and ABC, were slightly older, and underwent a mean of only 4.8 visits, which may account for the differences in improvements. Consistent with the current findings, a prospective study of participants with central vestibular disorders who underwent customized vestibular rehabilitation, consisting of gaze stabilization exercises, optokinetic training, habituation exercises, and static and dynamic gait and balance activities, showed improved postural control under various sensory conditions in 10 of 14 participants [33]. 

A recent retrospective study addressed the effectiveness of vestibular rehabilitation in 12 patients with idiopathic cerebellar ataxia and bilateral vestibulopathy (iCABV) [9]. Similar to the current study, treatment included customized gait and balance exercises, VOR gaze stabilization exercises, and habituation exercises over a course of six sessions, on average. [9]. Although their results showed improvements in all outcome measures, including ABC, DHI, DGI, and gait speed, after PT interventions, the only significant change was in time before a fall on condition four and total number of falls on the modified Clinical Test of Sensory Integration in Balance. 

The impacts of age and years from onset on the outcome measures was investigated. In our study, older adults (>60 years) improved to the same extent as the younger patients in the POMA, although they started at a lower score. One explanation is that the older patients in our study were sedentary prior to the interventions, resulting in muscle weakness and disuse disequilibrium which improved significantly as a result of therapy; however, we did not measure this. The subjects who had ataxia for >5 years did not show as much change as the participants with a more recent onset, suggesting that early intervention may be beneficial for these patients.

The participants with central or peripheral vestibular involvement (28/32 subjects) improved to a greater extent on the DGI than those with ataxia only, which was not expected. Possibly, the prescribed habituation and gaze stabilization exercises may have improved DGI scores because they also decreased dizziness. It also suggests that vestibular exercises are beneficial in patients with degenerative cerebellar disorders who are still ambulatory. These exercises may not be needed in patients who are not ambulatory unless they have significant dizziness. 

### Limitations

Retrospective studies have several well-known limitations. There is a higher likelihood of investigator bias affecting the results. Additionally, their results can be confounded by missing data. In particular, the Dynamic Visual Acuity [35] and Motion Sensitivity Tests [36,37] were not performed on all patients. These tests might have revealed more information on why patients with central and peripheral vestibular disorders improved to a greater extent in fall risk than those with imbalance only. The small sample size of patients with vestibular hypofunction (n = 8) is a limitation and shows that further research is needed with more subjects. Follow-up testing to evaluate whether participants maintained their improvements was not performed on all patients; therefore, those results were not included in this paper. Further research is needed to confirm that this type of rehabilitation is beneficial. Finally, compliance with HEP was not consistently recorded in a written calendar, it was only verbally discussed and documented in the medical record; therefore, we could not determine its effect on the outcomes.

## 5. Conclusions

This study focused on individuals with degenerative cerebellar ataxia who participated in a comprehensive exercise program including challenging gait and balance exercises as well as habituation and gaze stabilization exercises. Both objectives were addressed: (1) the three subgroups improved in balance confidence, fall risk, and sensory integration and (2) improvements were not affected by age or time from onset. It is noteworthy that more than 60% of our patients had central and/or peripheral vestibular impairments, suggesting that patients with cerebellar degeneration should have vestibular testing and, if abnormal, be referred to therapists specializing in the treatment of vestibular disorders to achieve optimal benefits from rehabilitation.

## Figures and Tables

**Table 1 brainsci-13-01520-t001:** Participant demographics (means/SD/range/*p*-values) for all patients and by groups.

Variables	All Patients (n = 42)	Ataxia Only (n = 14)	Ataxia + Oculomotor (n = 20)	Ataxia + Oculomotor + Vestibular (n = 8)	*p*-Values (Group Differences ^†^)
Age (y)	54.5 (14.4) 18–78	50.6 (10.4) 30–63	57.0 (16.0) 18–76	55.4 (16.6) 21–78	0.46
Sex (M/F)	23/19	7/7	12/8	4/4	0.81
Comorbidities (n)	1.8 (1.4) 0–5	1.8 (1.4) 0–5	1.8 (1.5) 0–5	1.5 (1.8) 0–5	0.86
Time from onset (years)	5.7 (6.5) 1–34	6.4 (8.7) 1–34	5.0 (5.2) 1–20	6.4 (5.6) 1–15	0.78
Number of visits (n)	5.2 (2.6) 2–15	4.3 (1.4) 2–7	5.4 (3.0) 2–15	7.1 (2.2) 5–11	0.04 *

^†^ ANOVAs except chi-square test for gender * post-hoc independent samples *t*-test: Ataxia + Oculomotor + Vestibular received significantly (*p* = 0.009) more visits than Ataxia Only.

**Table 2 brainsci-13-01520-t002:** Summary of findings (means/SD/range/*p*-values) by groups.

Outcomes	All Patients (n= 42)	Ataxia Only (n =14)	Ataxia + Oculomotor (n = 20)	Ataxia + Oculomotor + Vestibular (n = 8)	*p*-Values (Main Effect of Time)	(Interaction Time x Group)
ABC (%) Pre	(n = 29) 50.3 (20.2)	(n = 11) 48.9 (21.9) (22.5–87.5)	(n = 12) 52.7 (18.0) (27.5–88.8)	(n = 6) 48.0 (24.2) (11.2–75.6)	<0.001	0.81
Post	66.8 (18.9)	64.7 (23.8) (25.0–97.8)	67.9 (17.6) (40.0–88.7)	68.6 (13.2) (52.2–90.3)		
DGI (/24) Pre	(n = 22) 14.6 (4.2)	(n = 8) 14.4 (2.4) (11–18)	(n = 10) 16.2 (3.9) (10–22)	(n = 4) 10.8 (6.0) (3–16)	<0.001	0.03 *
Post	19.1 (2.9)	17.9 (3.2) (13–22)	20.0 (3.0) (14–23)	19.0 (1.2) (18–20)		
POMA (/37) Pre	(n = 27) 23.1 (6.6)	(n = 9) 23.9 (5.4) (18–33)	(n = 13) 22.7 (7.8) (14–35)	(n = 5) 22.6 (6.8) (17–33)	<0.001	0.63
Post	31.0 (4.4)	30.7 (4.7) (20–36)	30.7 (5.0) (19–37)	32.2 (2.3) (30–35)		
SOT (/64–70 ^§^) Pre	(n = 33) 49.1 (11.5)	(n = 9) 51.3 (14.2) (23–70)	(n = 18) 50.9 (10.8) (36–71)	(n = 6) 40.3 (4.1) (36–45)	<0.001	0.15
Post	64.5 (11.8)	63.2 (15.1) (43–88)	65.7 (9.9) (51–82)	62.5 (13.4) (45–73)		

ABC: Activities-specific balance confidence scale; DGI: Dynamic gait index; POMA: Performance oriented mobility assessment; SOT: Sensory organization test. ^§^ SOT Cut-off by age: <60 = 70/100, 60–69 = 68.6/100, ≥70 = 64/100. * Post-hoc test (Fisher’s LSD): Ataxia Only vs. Ataxia + Oculomotor, *p* = 0.20; Ataxia Only vs. Ataxia + Oculomotor + Vestibular, *p* = 0.52; Ataxia + Oculomotor vs. Ataxia + Oculomotor + Vestibular, *p* = 0.10.

**Table 3 brainsci-13-01520-t003:** Summary findings by sub-groups (age and time from onset).

Outcomes	Age Group (< 65 vs ≥65) (n = 32, n = 10)	Time from Onset Group (≤5 Years vs > 5 Years) (n = 29, n = 13)	*p*-Values (Interaction Time x Group) Age Onset	
ABC (%) Pre	(n = 21, n = 8) 48.2 (22.3) vs 55.7 (12.7)	(n = 20, n = 9) 47.9 (22.7) vs 55.5 (12.6)	0.23	0.69
Post	67.1 (20.4) vs 66.0 (15.8)	63.6 (19.5) vs 73.9 (16.4)		
DGI (/24) Pre	(n = 18, n = 4) 14.3 (4.4) vs 15.5 (3.9)	(n = 12, n = 10) 14.3 (5.0) vs 14.8 (3.3)	0.51	0.61
Post	19.1 (2.9) vs 19.0 (3.4)	19.2 (3.2) vs 18.9 (2.8)		
POMA (/37) Pre	(n = 20, n = 7) 24.7 (6.5) vs 18.4 (4.6)	(n = 23, n = 4) 24.0 (6.7) vs 17.8 (2.2)	0.04 *	0.28
Post	31.4 (4.7) vs 29.7 (3.4)	31.4 (4.4) vs 28.2 (3.5)		
SOT (/64–70 ^§^) Pre	(n = 23, n = 10) 50.4 (12.4) vs 46.1 (9.0)	(n = 23, n = 10) 47.5 (11.8) vs 52.7 (10.6)	0.39	0.68
Post	64.7 (12.7) vs 63.8 (10.1)	62.4 (12.0) vs 69.2 (10.4)		

ABC: Activities-specific balance confidence scale; DGI: Dynamic gait index; POMA: Performance oriented mobility assessment; SOT: Sensory organization test. ^§^ SOT Cut-off by age: <60 = 70/100, 60–69 = 68.6/100, ≥70 = 64/100. * Post-hoc independent samples t-test for younger vs. older adults: Pre-POMA scores, *p* = 0.02; Post-POMA scores, *p* = 0.66.

## Data Availability

The (de-identified) data presented in this study are available on request from the corresponding author (LHG). The data are not publicly available due to privacy.

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
