# Peer review of "Effectiveness of Vestibular Rehabilitation for Patients with Degenerative Cerebellar Ataxia: A Retrospective Cohort Study"

_brainsci, 2023, doi:10.3390/brainsci13111520_

Round 1

Reviewer 1 Report

Comments and Suggestions for Authors

This is a nice study, with some inherent limitations. Cerebellar ataxia is a challenging group of disorders to treat and to study. Research is hampered by the small sample sizes, especially in the subgroup that includes patients who also have peripheral vestibular weakness. This problem of sample size, especially in one group with only 8 patients, should be noted as a limitation in the Discission section on Limitations of the Study. Also, the relatively greater improvement by patients in the group with peripheral vestibular disorders was probably due to resolution of symptoms caused by vestibular impairments. The authors should have noted that issue in the Discussion. The authors did note the inherent problems with retrospective studies. They were incorrect in suggesting that adding the Dizziness Handicap Inventory would have enhanced the study. The DHI is a poor scale, which conflates perceptual, emotional and functional domains. The literature  includes better scales of each of those domains, which they may want to use in future prospective research. They should remove the comment about the DHI in this study. Another limitation of the study is the lack of post-treatment follow-up due to the retrospective design. Therefore, we cannot know if the effects of intervention out-lasted treatment. The authors should mention this problem in the Limitations section, and they should soften their comments about the actual usefulness of therapy. Although their enthusiasm is commendable they should not overstate the case. 

Reviewer 2 Report

Comments and Suggestions for Authors

Thank you for the opportunity to review this study, my considerations about the manuscript are described below:

Title:

- Is it a randomized clinical trial? If yes, it needs to be described in the title, according to CONSORT recommendations.

Abstract:

- I believe that adding subtopics (Background, objectives, methods, results and conclusion) will help improve the writing of the text and its understanding by readers. I suggest including these subtopics in the abstract.

- Description of the type of study must be in the abstract. In addition to descriptions of the methodological stages of the study, for example: randomization, division of the groups, sample size in each group, the number of groups in which the study was composed, description of the intervention in each group, how many days per week, for how many weeks, and the duration of each session.

- In the results, the authors report that the volunteers improved in all measures analyzed, however, it is not known how this improvement was. This needs to be better described, for example. Was the improvement intra-group? or inter-groups?

- The conclusion of the study should clarify the findings better, does this result cover the three groups (only ataxia, ataxia and oculomotor abnormalities and ataxia and peripheral oculomotor and vestibular abnormalities)?

- Were there any of the groups in which the effect size was larger? If so, it should be mentioned, as it directs the clinical practice of physiotherapy on the topic.

Introduction:

- The introduction is well written, presents a logical sequence and explains the study problem well.

- In the third paragraph, the authors mention data from systematic reviews, however, they refer to these studies as “systemic reviews”. Please correct the term for systematic reviews, as recommended by PRISMA.

- The objectives of the study in the abstract and at the end of the introduction are divergent, I suggest to the authors the standardization of objectives throughout the study.

Methods:

- What is the design/type of the study?

- How was the sample recruited? Was there a calculation to estimate the sample size, or is it a convenience sample? The authors need to further detail how the sample for this study was recruited.

- What are the inclusion and exclusion criteria for the study? Despite some descriptions about the sample, these criteria are not clear in the study.

- Was there approval from the research ethics committee to carry out the study? If yes, it needs to be mentioned in the methods, including the approval opinion number.

- From what I see of the study, it is a database from a clinic or hospital in which the authors decided to publish these findings.

- However, although the authors mention that this study is a medical record review, it provides primary results, never before published, so, in fact, it is an intervention study, or a non-randomized clinical trial .

- In a review study, the objective is to describe the results of other studies, never to provide new data or observe the effectiveness of a particular treatment. The authors should further clarify the design/type and methods of the study.

- Still, there is a lack of data from the study, such as: were there sample losses? Did all volunteers perform all pre- and post-intervention tests? What is the intervention period? Was the intervention period the same for participants? How many sessions were held? How long is each session? All this information is extremely relevant for physical therapy clinical practice and needs to be described in the manuscript.

- I suggest a better description of the exercises used in the interventions with the addition of photos, this is very important for clinical practice, as they increase the chances of these exercises being replicated and this study being cited by other future studies on the topic.

- Were the exercises used in the intervention replicated from any study? If so, it needs to be mentioned, for example, some of the exercises used look like the hand-eye coordination exercises proposed by Herdman, so the authors need to mention where these exercises were replicated from.

Results:

- Why didn't the authors use p-values in the tables? I suggest including.

- Including individuals from the three groups in a group only according to sex, age and time since the beginning does not seem reasonable to me in relation to clinical practice. One justification for this is that subjects with vestibular dysfunction normally present the worst performances, so this is the production of a result with the presence of many biases in terms of sample characteristics.

Discussion:

- The discussion is very vague and superficial, the authors only mention that they found results similar to those of other studies, this is not discussion of the findings.

- I missed in the text a physiological or pathophysiological explanation of how the exercises used may have influenced the improvement of outcomes, this would bring greater theoretical-scientific solidity to the findings and especially the intervention and its exercises.

- I suggest adding to the limitations, the methodological limitations of this study (lack of randomization, allocation secrecy, blinding of the outcome assessors, control of some biases and non-registration of the intervention protocol), and emphasizing that its results cannot be generalized.

Conclusion:

- According to the authors, the objective of this study was: to address this knowledge gap by determining: 1) whether patients with degenerative cerebellar ataxia can improve in balance confidence, fall risk and sensory integration for balance with a customized plan of care including vestibular gauze stabilization and habituation exercises to address central or peripheral vestibular deficits; and 2) whether age, time from onset of gait instability, or gender affects rehabilitation outcomes.

However, the text of the conclusion does not respond the objectives, I suggest rewriting the conclusion.

Author Response

this is the response to reviewer 2

Reviewer 3 Report

Comments and Suggestions for Authors

The nature of the study (retrospective review of cases) unfortunately compromises the reproducibility of the methods used and the reliability of the results obtained. Furthermore, as the authors themselves say in the text, among the limitations of the study, not all methods were applied to the entire sample and this fact could alter the results. In any case, the sample is made up of a good number of patients, the study is well written, the results are interesting for their significance with respect to this rare disease which is difficult to manage in clinical practice and the outcomes used for the measures are standardized and well replicable despite the retrospective nature of the study. 

On line 166 the authors describe in detail the frequency of Vestibular Gaze Stabilization Exercises performed at home. I think this should be done for the other exercises too. 

Line 191/193: you mention the characteristics of the three different groups and describe central oculomotor deficits (smooth pursuit, saccades, spontaneous/direction changing gaze evoked nystagmus or  VOR cancellation) but you don’t give a definition of peripheral vestibular deficit. What criteria did you use to include patients in the third group? the result of the caloric test? Explain.

Line 209 revise the sentence (the…without the subject). In the same sentence you write that the three groups were different at baseline, but in what parameters? Please describe better and insert the symbol of significance in Table 2. 

Table2: in ABC, the score varies from 50.3 % to 66.8 %; where does the 55.5% change percentage come from? Also, in DGI, from 14.6 to 19.1 is a change of 20%. I don’t understand. Explain better in the results or correct. 

Indicate the significance of results in table 3 (*p < 0.05, **p < 0.01, ***p < 0.001).

In general I reiterate that a more in-depth description of the results obtained and their meaning is necessary.

Author Response

This is the response for reviewer 3

Round 2

Reviewer 2 Report

Comments and Suggestions for Authors

I congratulate the authors for the changes to the manuscript, practically all of my requests were met. I believe that the manuscript has become clearer to readers and is now likely to be accepted for publication.